# Exploitation of a New Nucleating Agent by Molecular Structure Modification of Aryl Phosphate and Its Effect on Application Properties of the Polypropylene

**DOI:** 10.3390/polym15244730

**Published:** 2023-12-17

**Authors:** Fuhua Lin, Mi Zhang, Tianjiao Zhao, Yanli Zhang, Dingyi Ning, Wenju Cui, Yingchun Li, Xinde Chen, Jun Luo

**Affiliations:** 1School of Traffic Engineering, Shanxi Vocational University of Engineering Science and Technology, Jinzhong 030619, China; 2School of Materials Science and Engineering, North University of China, Taiyuan 030024, China; liyingchun@nuc.edu.cn; 3Shanxi Province Institute of Chemical Industry (Co., Ltd.), Jinzhong 030621, China; 18120166132@163.com; 4School of Chemical Engineering and Technology, Taiyuan University of Science and Technology, Taiyuan 030024, China; s202221211044@stu.tyust.edu.cn (T.Z.); 2021051@tyust.edu.cn (Y.Z.); 202021070217@stu.tyust.edu.cn (D.N.); 202021050202@stu.tyust.edu.cn (W.C.); 5Key Laboratory of Renewable Energy, Chinese Academy of Sciences, Guangzhou 510640, China; chenxd@ms.giec.ac.cn; 6Guangzhou Fibre Product Testing and Research Institute, Guangzhou 510220, China

**Keywords:** PP, nucleating agent, crystallization behavior, optical property, mechanical property

## Abstract

In this work, a novel α-nucleating agent (NA) for polypropylene (PP) termed APAl-3C-12Li was prepared and evaluated compared with the commercially available type NA-21. For the synthesis of the organophosphate-type NA (APAl-3C), the -OH group of the acid part of NA-21 was substituted by the isopropoxy group. The structure of APAl-3C was analyzed by spectroscopy and element analysis, the results of which were consistent with the theoretical molecular formula. APAl-3C’s thermal stability was studied by differential scanning calorimetry (DSC) and thermogravimetry (TG), which showed only weak mass loss below 230 °C, meaning that it would not decompose during the processing of PP. The APAl-3C-12Li was used as a novel nucleating agent, studying its effects on crystallization, microstructure, mechanical and optical properties. Tests were performed in a PP random copolymer at different contents, in comparison to the commercial NA-21. The composite with 0.5 wt% APAl-3C-12Li has a similar crystallization temperature of 118.8 °C as with the addition of 0.5 wt% NA-21. An advantage is that the composite with the APAl-3C-12Li has a lower haze value of 9.3% than the counterpart with NA-21. This is due to the weaker polarity of APAl-3C-12Li after the introduction of methyl and better uniform dispersion in the PP matrix, resulting in stronger improvement of optical and mechanical properties.

## 1. Introduction

Polypropylene (PP) is an important class of thermoplastics which is widely used in a variety of applications in fields such as packaging materials, household appliances and automotive industries [1,2,3,4]. Furthermore, PP is gaining more and more attention due to its favorable advantages of low cost, rich raw material source, easy processing, chemical resistance and good solvent resistance [5,6,7]. However, the crystallization rate of PP is low, and the large-size crystal leads to the disadvantages of PP materials such as time-consuming molding, poor transparency, and poor low-temperature impact resistance [8,9,10]. When used in different end-use domains, good crystallization property and mechanical properties for PP are generally required during mold processing. From the industrial application point of view, PP can be modified by various methods to expand its range of application [11,12].

It is reported that the crystallization capacity of polymers is significantly increased in the presence of nucleating agents and one of the most attractive and most mentioned frequently belongs to an α-nucleating agent [13]. Incorporation of α-nucleating agents into the PP matrix can provide a new solution to improve the crystallization and mechanical properties in the PP industry, where the presence of α-nucleating agents can promote the ability to form crystal nuclei, increase the rate of heterogeneous nucleation and crystallization, and reduce the spherulite size of PP [14,15]. Furthermore, the mechanical and optical properties of PP, including impact strength, tensile strength, flexural modulus and haze value, can be improved by reducing the size of spherulites [16,17,18,19]. The traditional organic α-nucleating agents such as dibenzylidene sorbitol [20,21], aryl phosphates [22,23], carboxylates and rosin salts [24,25] have been developed to enhance the crystallization property of PP.

Aryl phosphate (AP), as one of high-performance α-nucleating agents, can significantly change the crystallization behavior and reduce the spherulite size of PP by their low loadings, thus reinforcing the crystallinity and crystallization rate of PP, and effectively promoting the mechanical properties of PP further. Many studies about the modification of PP by AP nucleating agents have been reported, focusing on raising the density of crystal nucleus, reducing the spherulite size, improving the optical performance, shortening the processing cycle, and enhancing the charge storage property and the mechanical properties [26,27,28]. Min G [29] et al. prepared composite nucleating agents using a silicon nanoparticle-modified sodium phosphate nucleating agent (NA-11) to improve the mechanical property and crystallization behavior of PP. The results showed that the mechanical properties of PP were significantly improved by adding a composite nucleating agent into the PP matrix. The modified nucleating agents had a significant effect on enhancing the stiffness of PP with the addition of 0.2 wt%, which was similar to using NA-11 at such addition. Zhang [30] et al. investigated the nucleating ability of NA-21 on PP through the isothermal crystallization kinetics of PP by DSC; the results indicated that the addition of NA-21 in PP markedly shortened the semi-crystallization time and reduced the crystallization activation energy during isothermal crystallization.

In terms of APs, they underwent three generations of structural changes, all of which are widely used in PP around the world. Taking the nucleating agent products (ADEKA Corporation, Japan) as an example, the first generation of sodium monophenol phosphate salt (sodium bis (p-tert-butylphenoxy) phosphate) without bridged methylene (commercial product name: NA-10) [31] has low cost, high melting temperature and is mostly used for stiffness modification for PP. The second generation is sodium 2,2-methylene-bis (4,6-di-tert-butylphenyl) phosphate (commercial product name: NA-11) [32]. It has good thermal stability and thermal decomposition temperature of more than 400 °C, and the nucleation efficiency is higher than that of NA-10. Nevertheless, it is not well dispersed in the polymer matrix and has limited ability to improve the transparency of PP. The third generation is a compound nucleating agent composed of aluminum phosphate basic salt as the main component and lauric acid salt as the ligand (commercial product name: NA-21) [33]. The main composition of NA-21 is bis [2,2′-methylene-bis(4,6-di-tert-butylphenoxy) phosphoric acid] hydroxyl aluminum (APAl-OH), which has strong nucleation ability and has the effect of increasing stiffness. The structure of three generations AP nucleating agents is shown in Figure 1.

Based on the structure of AP nucleating agents shown in Figure 1, it is found that the number of -CH_3_ is increasing with the upgrade, which is speculated to be the key role of -CH_3_ on nucleation. To verify this hypothesis, we prepare a new material of APAl-3C by replacing hydroxyl in APAl-OH as the main composition of NA-21 with isopropoxy. On the other hand, the addition of -CH_3_ can improve the interfacial compatibility between APAl-3C and polymer matrix. Herein, we first construct a novel nucleating agent (APAl-3C-12Li) compounded by lithium lauratie (12Li) and APAl-3C, where APAl-3C is synthesized based on 2,2′-methylene bis(4,6-di-tert-butyl phenoxy) phosphate (APOH) and aluminum isopropoxide (Al(OiPr)_3_) through co-precipitation methods. The PP composites with nucleating agent APAl-3C-12Li are prepared using melt blending. The crystallization behavior as well as the optical and mechanical properties in PP are systematically studied in detail and then compared with those of commercial nucleating agent NA-21. In addition, the effect of the isopropyl group in the structure of nucleating agent APAl-3C-12Li on the crystallization properties of PP composites is also examined by differential scanning calorimetry (DSC) and a polarizing microscope (POM).

## 2. Materials and Methods

### 2.1. Materials

Polypropylene (PP) was provided by Yanshan Petrochemical Co., Ltd. China. Aluminum isopropoxide (Al(OiPr)_3_), lithium laurate (12Li) and 2,2′-methylene bis(4,6-di-tert-butyl phenoxy) phosphate (APOH) was purchased from Shanghai McLean Biochemical Technology Co., Ltd. China. Toluene was obtained from Shanghai Lingfeng Chemical Co., Ltd., Shanghai, China. All reagents mentioned were of analytical grade except PP. NA-21 was provided by Milliken Co., Ltd., Milliken*,* CO, USA.

### 2.2. Preparation of APAl-3C

APOH and Al(OiPr)_3_ set at 2:1 (molar ratio) were dispersed fully in a 100 mL toluene, then the mixture solution was added to the round bottom flask and heated for 2 h while stirring. APAl-3C was filtered to remove the solvent, washed with toluene and dried in an oven at 105 °C for 12 h. The corresponding synthesis chart of APAl-3C is shown in Figure 2.

### 2.3. Preparation of the Nucleating Agent

The powder samples of APAl-3C and lithium laurate (12Li) were mixed evenly at room temperature in a hopper machine, and the mass ratio was about 6:4. The mixed nucleating agent was named APAl-3C-12Li.

### 2.4. Preparation of the PP Composites

PP was melt-blended with different contents of nucleating agent APAl-3C-12Li or NA-21 in a co-rotating twin screw extruder (TSH-25, Chuangbo Machinery Equipment Co., Ltd., Nanjing, China). The temperatures of the four sections were 175 °C, 180 °C, 185 °C, and 200 °C, respectively, and the screw speed was set to 300 r/min. The mixture after granulation was injected into a suitable sample at 200 °C by a MA1200/370 micro-injection molding machine (Haitian Plastic Group Co., Ltd., Ningbo, China). The formula of the PP composites is shown in Table 1.

### 2.5. Measurement and Characterization

Fourier transform infrared spectra (FTIR, Spectrum 100, Perkin Elmer Corporation, Waltham, MA, USA) were recorded at a resolution of 4 cm^−1^ in the range of 4000–700 cm^−1^ to determine the structure of APAl-3C.

A nuclear magnetic resonance hydrogen spectrogram (^1^H NMR, Avance 500 MHz, Bruker Corporation, Ettlingen, Germany) was recorded employing a nuclear magnetic resonance spectrometer and operated at a frequency of 500 MHz at room temperature. The sample was dissolved in deuterated methanol as a solvent to prepare solutions of 5 *w*/*v*%, and the sweep frequency width was 6 kHz.

Element analyzer (EA, Vario EL cube, Elementar Corporation, Langenselbold, Germany) was used to test the content of C and H. The temperatures of oxidation furnace and reduction furnace were set as 1150 °C and 850 °C, and the flow rates of He and O_2_ were set as 230 mL/min and 16 mL/min, respectively.

Ultraviolet-visible spectrophotometer (UV-vis, Lambda 650, Perkin Elmer Corporation, USA) was used to test the content of P and Al. The preliminary treatment of samples was in accordance with the wet digestion method.

The contact angle meter (SL200B goniometer, Kino, Pawai, PL, USA) was used to investigate contact angle test by the sessile drop method with water as a solvent after the samples were pressed into tablets.

Thermogravimetric analysis (TG, TGA1, Mettler Toledo Corporation, Columbus, OH, USA) was carried out at 10 °C/min from 50 °C to 800 °C under N_2_ atmosphere. The mass of the samples in the ceramic crucible was maintained between 5 and 10 mg.

Differential scanning calorimeter measurements (DSC, DSC1, Mettler Toledo Corporation, Columbus, OH, USA) were used to examine the non-isothermal crystallization behavior of the PP composites under N_2_ atmosphere. The samples were first heated from 25 °C to 210 °C at a heating rate of 50 °C/min and maintained at 210 °C for 5 min to eliminate thermal history. Then, the samples were cooled from 210 °C to 25 °C at a cooling rate of 10 °C/min to obtain the crystallization curve. The melting curve was obtained by re-heating to 210 °C at a rate of 10 °C/min. The sample was weighted in the range of 5~6 mg.

A polarizing microscope (POM, Leica DM 2700, Wetzlar, Germany) has often been applied to observe the crystal morphology of PP composites. The sample was heated from 30 °C to 210 °C at a rate of 50 °C and kept for 5 min to eliminate thermal history. After that, the samples were rapidly cooled to 135 °C and then kept at this temperature to observe the crystallization process until the crystallization process was completed.

A universal testing machine (M10, Instron, Norwood, MA, USA) was used for the tensile strength and the flexural modulus test of PP composite materials. The crosshead speed was set at 5 mm/min during the tensile strength test according to the GB/T1040.2-2006. The flexural modulus test used a three-point bending test mode, and the bending rate was set at 2 mm/min according to GB/T 9341-2008. An impact testing machine (GT-7045-HML, Gaotie, Taiwan, China) was used for impact strength test. The crosshead speed was set at 5 mm/min during the test according to GB/T 1843-2008. The test temperature was 25 ± 2 °C, and each sample was tested at least five times to obtain an average value with standard deviation.

Photoelectric Colorimeter (WGT-2S, Shanghai INESA Physico-Optical Instrument Co., Ltd., Shanghai, China) was used to test the haze and clarity value of PP composites with reference to GB/T2410-2008 with the sample size of (50 mm ± 2 mm) × (50 mm ± 2 mm). The values were averaged over five measurements.

A Vicat softening point testing machine (WKW-300B, Intelligent Machinery Equipment Co., Ltd., Changchun, China) was used to obtain the Vicat softening temperature (VST) value of PP composites with reference to GB/T1633-2000. The sample size was (10 mm ± 1 mm) × (10 mm ± 1 mm) × (5 mm ± 1 mm) and the values were the average value of the data after five measurements.

## 3. Result and Discussion

### 3.1. Characterization of APAl-3C

Figure 3a presents the FTIR spectra of Al(OiPr)_3_, APOH and APAl-3C. Regarding the spectrum of APAl-3C, the peak at about 3500 cm^−1^ attributed to the characteristic peak of -OH could not be observed [34]. Moreover, the peak at 1020 cm^−1^ in the spectrum of APOH is related to the stretching vibration of P-OH, which does not appear in spectrum of APAl-3C. This phenomenon proves that the atom hydrogen in APOH is replaced by isopropyl in Al(OiPr)_3_ in the reaction process of APAl-3C. There are many characteristic absorption peaks in the range of 1600–1500 cm^−1^, which are ascribed to the characteristic absorption peaks of benzene ring [35]. The absorption peaks at about 1365 and 1395 cm^−1^ are assigned to -CH_3_ of the tert-butyl functional group and the isopropyl functional group in the structure of APAl-3C [36], where the introduction of the -CH_3_ group does not influence the peak position, but the area ratio of the two peaks changes. The stretching vibrations of P-O-Al and P=O-Al are detected at approximately 1230 and 1080 cm^−1^, respectively [37]. The FTIR results are consistent with the characteristic absorption peaks of APAl-3C. Figure 3b represents the 400 MHz ^1^H-NMR spectrum measured at room temperature for APAl-3C dissolved in 5 wt% CD_4_O to reconfirm the structure of APAl-3C, showing the attribution of chemical shifts of hydrogen atoms at different functional group positions in the APAl-3C structure as exhibited. The chemical shifts at 1.46 and 1.33 ppm are assigned to the H band of the tert-butyl group on the benzene ring (‘3’ and ‘4’ -CH_3_) as reported in literature [38], respectively. The proton displacements on the benzene ring in the structure of APAl-3C appear at 7.31 ppm and 7.21 ppm (‘1’ and ‘2’ C-H) [39]. The peak position of the methylene group connected with two benzene ring structures appears at 3.81 ppm (‘7’ -CH_2_-). The methylene group in the middle of the isopropyl group appears at 3.63 ppm (‘5’ -CH-) [40], connected with two methyl groups, so the peak appears as multiple splitting peaks. The terminal methyl hydrogen C-CH_3_ of the isopropyl group appears at 1.18 ppm (‘6’ -CH_3_). The positions of hydrogen atoms in the APAl-3C structure are in accordance with the results of FTIR. The element content of APAl-3C shows that the mass fractions of C, H, P and Al are 68.34, 8.68, 5.86 and 2.55%, respectively. It is calculated that the molar ratio of C:H:P:Al is about 61.0:91:1.9:1.0 (calculated based on Al content as normalization). The measured values of each element are basically close to the theoretical content, which is consistent with the theoretical molecular formula of APAl-3C. The results of element, FTIR and ^1^H-NMR corroborate the successful synthesis for APAl-3C.

The TG and DTG curves of APAl-3C ranging from 35 to 800 °C in a N_2_ atmosphere are exhibited in Figure 4a. In the case of APAl-3C, the decomposition process includes two mass loss stages. The first mass loss is mainly due to the disappearance of oxygen-containing groups, which occurs in the range of 300 to 450 °C [41], where the weight loss is about 53% and the peak temperature is 375 °C. The second decomposition stage between 450 and 800 °C is because of the disintegration of the carbon skeleton of APAl-3C [42], where the weight loss is about 21% and the peak temperature is 529 °C. The temperature at which a 5% weight loss occurs is defined as the initial decomposition temperature (T_5wt%_) of APAl-3C. T_5wt%_ is 245 °C, which is higher than the molding temperature of PP. In general, the prepared APAl-3C exhibits good thermal stability, which is suitable for PP processing and usage. Figure 4b represents the DSC curve of APAl-3C measured ranging from 35 to 200 °C in a N_2_ atmosphere. It can be seen that APAl-3C has an endothermic peak at 148 °C. Figure 4c represents the contact angles of APAl-OH and APAl-3C measured at the same time. The hydrophobicity of APAl-3C is 89.7°, higher than 77.3° of APAl-OH because of the presence of -CH_3_.

### 3.2. Non-Isothermal Crystallization Behavior of the PP Composites

The non-isothermal crystallization behavior studies are carried out in a N_2_ atmosphere as shown in Figure 5a,b. For the cases of non-isothermal crystallization, crystallization parameters such as crystallization peak temperature (T_c,p_), crystallization enthalpy (ΔH_c_) and the T_c,p_ difference between the PP composites and pure PP (ΔT_c,p_) are shown in Table 2. One can observe that both the PP composites and pure PP have only one crystallization peak, which certifies that the presence of nucleating agent APAl-3C-12Li or NA-21 has no effect on the shape of PP crystallization curves. It can be seen that the value of T_c,p_ generally increases with the increasing content of APAl-3C-12Li or NA-21 loaded in a PP matrix. The T_c,p_ of pure PP is 107.6 °C. When the addition amount of APAl-3C-12Li is 0.5 wt%, the T_c,p_ of PP/AP-3 composite reaches 118.8 °C and the ΔT_c,p_ is 11.2 °C. When the addition amount of NA-21 is 0.5 wt%, the PP/NA-3 composite reaches the largest value, increasing to 118.9 °C. The crystallization results show that the addition of the nucleating agent (APAl-3C-12Li or NA-21) can move the T_c,p_ value to the high-temperature region, promoting the PP composite crystallization at higher temperatures. The literature cites that the higher the T_c,p_, the stronger the nucleation ability and the faster the crystallization rate [43], indicating the presence of APAl-3C-12Li or NA-21 as an efficient nucleating agent. The ΔH_c_ value of PP/AP composites with the incorporation of APAl-3C-12Li is always higher than that of PP/NA composites with the incorporation of NA-21 under the same addition amount, suggesting the superior nucleating ability of APAl-3C-12Li, which rapidly accelerates the crystallization of PP.

The overall melt behavior studies are carried out in a N_2_ atmosphere as shown in Figure 5c,d. For the cases of non-isothermal crystallization, the melt parameters such as melt peak temperature (T_m,p_), melt enthalpy (ΔH_m_) and temperature difference of melt peak temperature between the PP composites and pure PP (ΔT_m,p_) are shown in Table 3. As shown in Figure 5c,d, pure PP has a long and wide melting peak at 146.7 °C. When adding the APAl-3C-12Li nucleating agent, the value of T_m,p_ gradually increases with the increasing content. When the amount is 0.5 wt%, the value of T_m,p_ for PP/AP-3 is 150.1 °C, 3.4 °C higher than that of pure PP, implying that the APAl-3C-12Li nucleating agent can increase the T_m,p_ value and make the PP crystal more perfect and dense. It is worth mentioning that PP/NA composites display a trend of increasing first and then decreasing under the presence of the NA-21 nucleating agent. This may be due to the uneven dispersion of high-content NA-21 in the PP matrix, thus leading to the imperfect crystal structure of PP and more crystal defects. This phenomenon does not occur in PP/AP composites, which is due to the introduction of non-polar functional groups in the APAl-3C-12Li structure. The incorporation of the two nucleating agents increases the crystallization and melting temperatures of PP composites, which represents an increase in the degree of crystallization ordering, demonstrating that the two nucleating agents play an important part in the crystallization properties of PP [44].

### 3.3. POM of the PP Composites

As we all know, a high clarity of samples is associated with the reduction in spherulite size [45]. The spherulite size of PP composites is investigated by POM under the condition of 135 °C in Figure 6. The pure PP exhibits a small amount of spherulites with a diameter of above 100 μm shown in Figure 6a. In the image of the PP/AP-1 composite with the incorporation of 0.1 wt% APAl-3C-12Li as shown in Figure 6b, more nucleation sites are formed; thus, a much smaller spherulite size is observed, the number of crystal nuclei dramatically increases and numerous fine crystals appear in about 1 min, which demonstrates that APAl-3C-12Li indeed serves as an effective nucleating agent in the PP matrix, increasing the density and reducing the size of spherulites. As shown in Figure 6c, compared with pure PP, the PP/NA-1 composite with the incorporation of 0.1 wt% NA-21 induces a large number of nuclei at the same time. It is noteworthy that the individual spherulites are not visible in the PP/AP-1 composite after adding 0.1 wt% of APAl-3C-12Li. The combination of POM and DSC results provide a conclusion that APAl-3C-12Li is more effective in nucleation activity than NA-21.

### 3.4. Mechanical Properties of the PP Composites

One purpose of preparing a new nucleating agent is to improve the interfacial interaction between nucleating agent APAl-3C-12Li and the PP matrix, thus improving the dispersion ability, which could in turn enhance the mechanical properties of PP composites [46]. The mechanical property data of PP composites are summarized in Figure 7 and Table 4, including tensile strength, flexural modulus and impact strength, which evaluate how the nucleating agent APAl-3C-12Li affects the mechanical properties of PP composites at different contents. In Figure 7a, pure PP has a low tensile strength value of 25.9 MPa owing to intrinsic rigidity [47]. Since the addition of APAl-3C-12Li to the PP matrix induces the tensile strength of PP to increase gradually, as a consequence of the substitution of isopropoxy, it promotes the dispersion between APAl-3C-12Li and the PP matrix, leading to increasing the interaction between PP molecular chains. On the other hand, the heterogeneous nucleation of APAl-3C-12Li in the PP matrix can effectively reduce the size of spherulites, improve the crystallinity of PP composites, and make the regular arrangement of PP molecular chains lead to the restriction of chain segment movement, causing the PP composites to become not easily deforming [48]. The maximum data are recorded about 28.3 MPa for PP/AP-1 after adding 0.1 wt% of APAl-3C-12Li. While adding more APAl-3C-12Li, the aggregation of APAl-3C-12Li exists as a defect in the PP matrix, resulting in the decrease in tensile strength. However, at a lower addition amount, the value of tensile strength for the PP/AP-3 composite is still higher than that of PP/NA-3 when incorporating the NA-21 nucleating agent. This is because PP/AP composites cannot easily produce cracks and defects under tensile processing, which is mainly attributed to the fact that APAl-3C-12Li has weaker polarity and stronger hydrophobicity than NA-21, which has better compatibility with PP and stronger force, thus having higher tensile strength.

In terms of flexural modulus, it is observed that the values for PP/AP composites are progressively increased with the presence of APAl-3C-12Li as revealed in Figure 7b. It is found that the flexural modulus of pure PP is approximately 952.7 MPa compared with 1058.4, 1084.1 and 1105.8 MPa of PP/AP-1, PP/AP-2, and PP/AP-3, respectively. However, it is worth noting that the flexural modulus of PP composites can be effectively adjusted by APAl-3C-12Li and the NA-21 nucleating agent, and it increases rapidly with the increase in the addition amount. When the addition amount is 0.5 wt%, the flexural modulus values of PP/AP-3 and PP/NA-3 reach the maximum, which are 1105.8 MPa and 1108.5 MPa, respectively, 16.1% and 16.3% higher than that of pure PP. This is because both APAl-3C-12Li and NA-21 can refine the spherulite size, thereby reducing the interfacial tension between the spherulites. In addition, the nucleating agents have a V-shaped structure conformation, which matches the PP molecular chain. It can restrain the movement of PP molecular chains and reduce the movement ability of PP chains [49], leading to making the arrangement of PP crystals more regular and orderly, and making the PP composites have higher rigidity.

Regarding the aspect of impact strength, for PP/AP and PP/NA composites, it is observed that the values are similar to that of the tensile strength of PP/AP composites, which has a trend of increasing sharply and then decreasing slowly with the increasing amount. The addition amount of the two kinds of nucleating agents to achieve the maximum impact strength is 0.1 wt%. After the addition of 0.1 wt% nucleating agents, the impact strength of the PP/AP-1 composite is 5.1 kJ/m^2^, which is higher than that of the PP/NA-1 composite (5.0 kJ/m^2^). With the increasing content of the nucleating agent, the impact strength of PP composites decreases slowly. Under the same addition amount, the impact strength of PP/AP composites is higher than that of PP/NA composites. These mechanical results demonstrate the fact that the addition of APAl-3C-12Li in the PP matrix promotes the compatibility and decreases the content of the amorphous region content, which improves the stiffness, whereas the toughness is almost unchanged, consequently leading to a rigid-tough balance.

### 3.5. Optical Property of the PP Composites

Clarity and haze are another important way to evaluate the nucleation ability of nucleating agents, and also an important index for commercial applications of polymers. Therefore, the clarity and haze of PP composites are measured to reflect the optical properties displayed in Figure 8. As we can see, the presence of nucleating agents APAl-3C-12Li and NA-21 has little effect on the clarity of PP. However, with the increase in the nucleating agent addition amount, the haze value of PP composites gradually decreases, which displays relative excellent haze properties, where the value of haze for pure PP is approximately 36.0% compared with 9.3% and 10.2% of PP/AP-3 with the content of 0.5 wt% APAl-3C-12Li and PP/NA-3 with the content of 0.5 wt% NA-21, respectively, which are 74.2% and 71.7% lower than that of the pure PP. Obviously, both APAl-3C-12Li and NA-21 have the ability to improve the optical properties of PP, which is attributed to the fact that the addition of APAl-3C-12Li and NA-21 can play a role in refining spherulites. After adding APAl-3C-12Li and NA-21 nucleating agents, the crystallization rate is increased and the size of the spherulites becomes smaller. While the size of spherulites is smaller than the size of visible light, the light passing through does not cause light loss due to the scattering and refraction of spherulites, thereby reducing the haze value and achieving high transparency of PP composites [50]. The haze value of the PP/AP-3 composite is lower than that of PP/NA-3 when the addition amount is higher than 0.1 wt%, which indicates that APAl-3C-12Li has greater contribution in improving the optical properties of PP, which is due to the superior compatibility and dispersion between APAl-3C-12Li and PP compared to that of NA-21. The observations of optical properties are in agreement with the results of POM performance.

### 3.6. VST of the PP Composites

Figure 9 exhibits the effect of APAl-3C-12Li on the heat resistance of the PP matrix under different addition amounts. It can be seen that for PP/AP composites, the VST value increases rapidly and then tends to be stable with the increase in APAl-3C-12Li amount. When the addition amount of APAl-3C-12Li is 0.1 wt%, the VST value reaches the maximum of 78.3 °C, 3.7 °C higher than that of pure PP. As the addition amount continues increasing, the VST value of PP/AP composites remains basically stable and does not change greatly with the addition amount of APAl-3C-12Li. For PP/NA composites, the change trend of the VST value is different from that of PP/AP composites. With the increasing amount of NA-21, the VST value increases slowly first and then decreases. When the addition amount of NA-21 is 0.3 wt%, it reaches the maximum of 79.1%. When the addition amount is 0.5 wt%, the VST value decreases rapidly to 77.0 °C. The addition of APAl-3C-12Li increases the crystallization temperature of PP composites, which indicates that the ordering degree of crystallization is improved, the density of the spherulites is improved after refinement, and the resistance to thermal deformation is enhanced [51]. It is worth mentioning that at low addition amount, the thermal deformation resistance of PP/AP composites increases rapidly. This may be because the introduction of isopropyl groups makes APAl-3C-12Li have good compatibility with PP matrix and reduces the polarity of the APAl-3C-12Li surface, so that it is not easy to agglomerate due to the increase in the addition amount, thus promoting its ability to resist thermal deformation, which is consistent with the decrease in the VST value of NA-21 at a high addition amount.

## 4. Conclusions

The results of SEM, FTIR and ^1^H-NMR indicate that the new nucleating agent APAl-3C-12Li with 5.56 wt% of P and 2.51 wt% of Al is successfully synthesized. The prepared APAl-3C-12Li exhibits good thermal stability and causes the improvement in mechanical and crystallization properties, which is suitable for PP processing and using application. The nucleation ability of APAl-3C-12Li nucleating agents on PP is analyzed by DSC and POM and compared with the ability of NA-21. Incorporating nucleating agent APAl-3C-12Li into the PP matrix can increase the crystallization temperatures and accelerate the crystallization rate of PP, indicating that the degree of ordering of the spherulites and the density of the spherulites are improved after refinement. The ΔHc value of PP/AP composites with APAl-3C-12Li is always higher than that of PP/NA composites with NA-21 under the same addition amount, suggesting the superior nucleating ability of APAl-3C-12Li, which rapidly accelerates the crystallization of PP. This can be explained by the introduction of non-polar functional groups in the APAl-3C-12Li structure. The mechanical analysis shows that PP/AP composites display remarkable enhancement in terms of tensile strength and impact strength compared with that of pure PP. The presence of non-polar functional groups in the APAl-3C-12Li structure promotes compatibility with the PP matrix, which makes PP/AP composites a preferable impact strength in comparation with PP/NA composites. Furthermore, the noteworthy fact is that the PP/AP-3 composite with 0.5 wt% of APAl-3C-12Li has the lowest haze value of 9.3%, which is lesser than that of the PP/NA-3 composite with 0.5 wt% of NA-21.

## Figures and Tables

**Figure 1 polymers-15-04730-f001:**
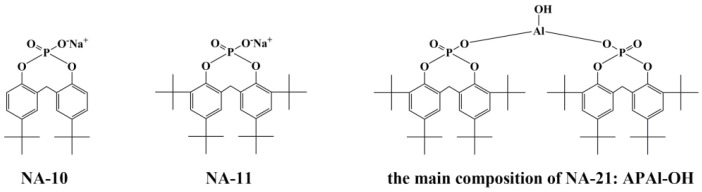
The structure of three generations for AP nucleating agents.

**Figure 2 polymers-15-04730-f002:**
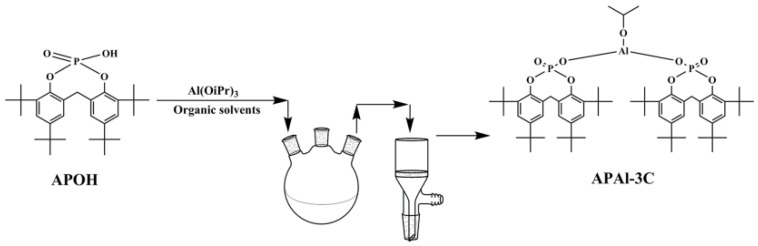
The synthesis chart of APAl-3C.

**Figure 3 polymers-15-04730-f003:**
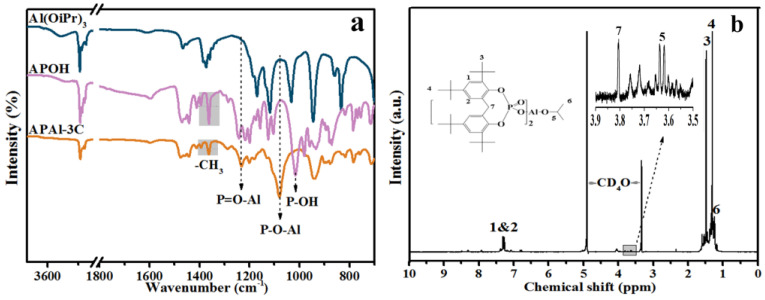
FTIR (**a**) and ^1^H-NMR (**b**) of APAl-3C.

**Figure 4 polymers-15-04730-f004:**
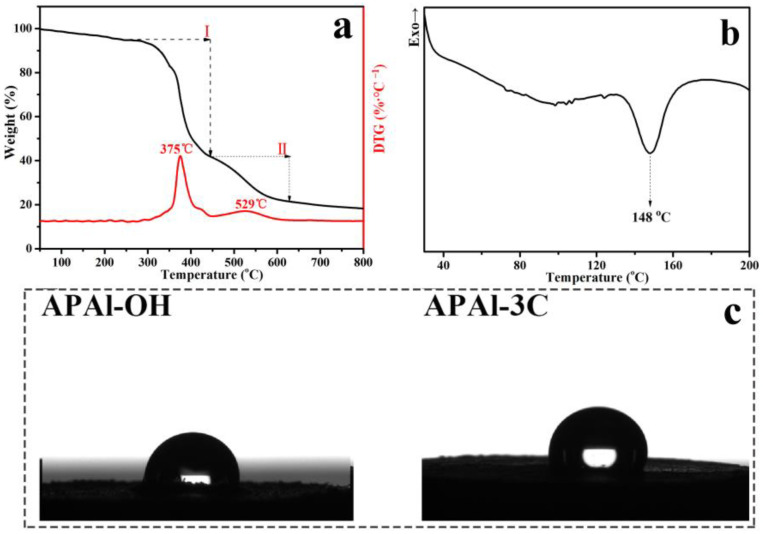
The TG and DTG curve of APAl3C (**a**), the DSC curve of APAl-3C (**b**) and contact angles of APAl-OH and APAl-3C (**c**).

**Figure 5 polymers-15-04730-f005:**
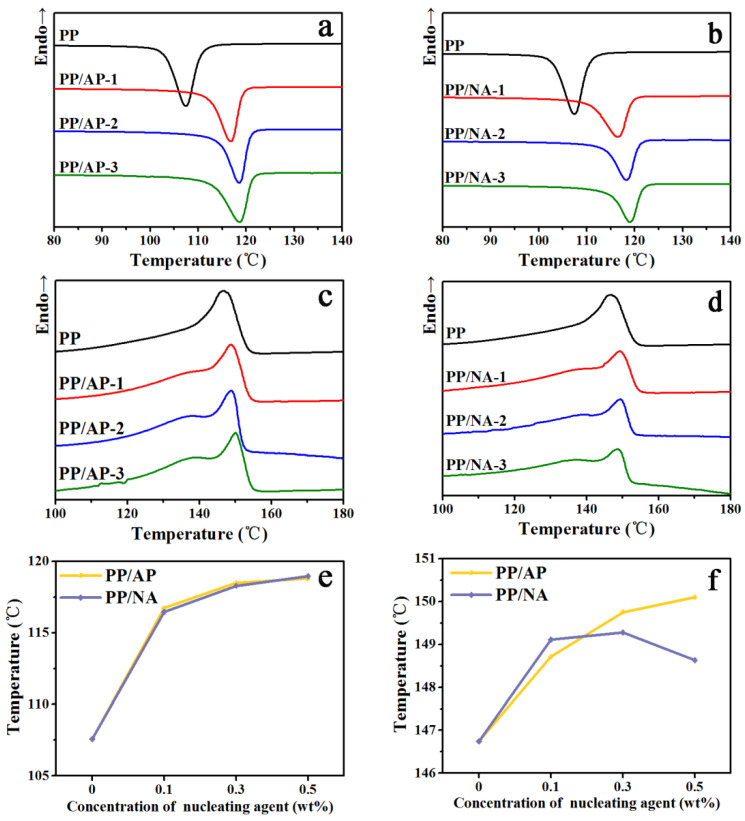
Crystallization and melting behavior of PP composites during a cooling rate of 10 °C/min under a N_2_ atmosphere: crystallization curve of PP/AP (**a**) and PP/NA (**b**); melting curve of PP/AP (**c**) and PP/NA (**d**); comparison curve of crystallization peak temperature (**e**) and melting peak temperature (**f**).

**Figure 6 polymers-15-04730-f006:**
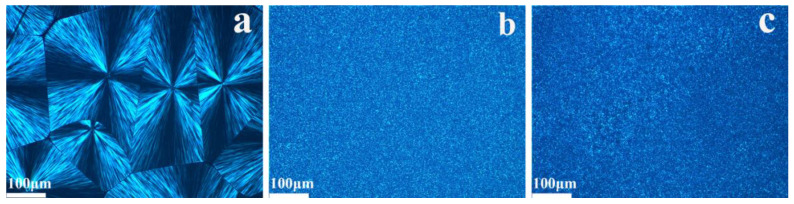
POM morphology of the PP composites at 135 °C: pure PP (**a**); PP/AP-1 (**b**) and PP/NA-1 (**c**).

**Figure 7 polymers-15-04730-f007:**
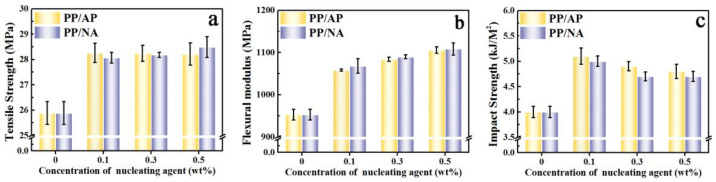
The mechanical properties of the PP composites: tensile strength (**a**), flexural modules (**b**) and impact strength (**c**).

**Figure 8 polymers-15-04730-f008:**
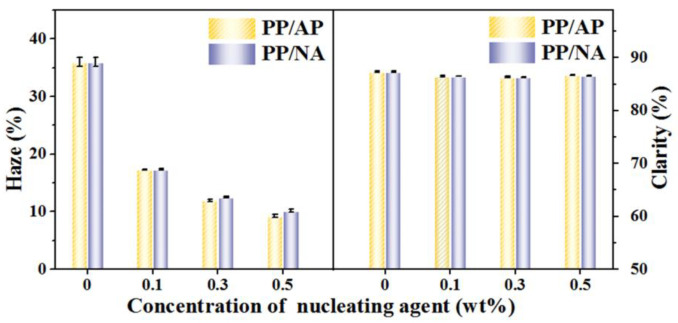
The optical property of the PP composites.

**Figure 9 polymers-15-04730-f009:**
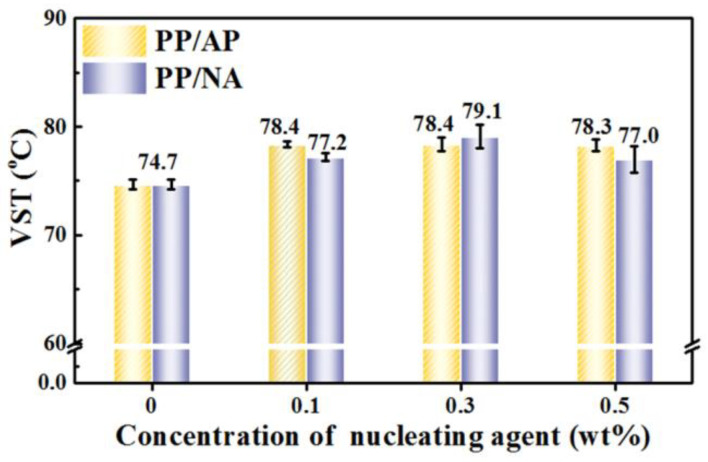
The VST value of the PP composites.

**Table 1 polymers-15-04730-t001:** The formula of the PP composites.

Samples	PP/g	APAl-3C-12Li/g	NA-21/g
PP	1000	0	0
PP/AP-1	999	1	0
PP/AP-2	997	3	0
PP/AP-3	995	5	0
PP/NA-1	999	0	1
PP/NA-2	997	0	3
PP/NA-3	995	0	5

**Table 2 polymers-15-04730-t002:** The crystallization properties of the PP composites.

Samples	*T*_c,p_ (°C)	Δ*T*_c,p_ (°C)	Δ*H*_c_ (J/g)	Samples	*T*_c,p_ (°C)	Δ*T*_c,p_ (°C)	Δ*H*_c_ (J/g)
PP	107.6	-	105.6	-	-	-	-
PP/AP-1	116.7	9.1	99.1	PP/NA-1	116.5	8.9	105.0
PP/AP-2	118.5	10.9	105.0	PP/NA-2	118.3	10.7	102.0
PP/AP-3	118.8	11.2	110.4	PP/NA-3	118.9	11.3	93.9

**Table 3 polymers-15-04730-t003:** The melting properties of the PP composites.

Samples	*T*_m,p_ (°C)	Δ*T*_m,p_ (°C)	Δ*H*_m_ (J/g)	Samples	*T*_m,p_ (°C)	Δ*T*_m,p_ (°C)	Δ*H*_m_ (J/g)
PP	146.7	-	92.3	-	-	-	-
PP/AP-1	148.7	2.0	98.2	PP/NA-1	149.1	2.4	105.9
PP/AP-2	149.8	3.1	106.7	PP/NA-2	149.3	2.6	99.7
PP/AP-3	150.1	3.4	111.6	PP/NA-3	148.6	1.9	96.4

**Table 4 polymers-15-04730-t004:** The mechanical dates of the PP composites.

Samples	Tensile Strength (MPa)	Flexural Modulus (MPa)	Impact Strength (kJ/m^2^)
PP	25.9	952.7	4.0
PP/AP-1	28.3	1058.4	5.1
PP/AP-2	28.2	1084.1	4.9
PP/AP-3	28.2	1105.8	4.8
PP/NA-1	28.1	1068.2	5.0
PP/NA-2	28.2	1089.8	4.7
PP/NA-3	28.5	1108.5	4.7

## Data Availability

The data presented in this study are available on request from the corresponding author.

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
