# Peer review of "Exploitation of a New Nucleating Agent by Molecular Structure Modification of Aryl Phosphate and Its Effect on Application Properties of the Polypropylene"

_polymers, 2023, doi:10.3390/polym15244730_

Round 1

Reviewer 1 Report

Comments and Suggestions for Authors

This is a very nice study of a novel organophosphate-type nucleating agent for iPP. The paper, however, suffers from a number of detail issues and mostly from poor language quality (see below). 

As an example, please find here a suggested better version of the abstract (remember, abbreviations must be spelt out at their first use - and the abstract IS the first use of many!):

"In this work, a novel α-nucleating agent (NA) for isotactic polypropylene (iPP) termed APAl-3C was prepared and evaluated. For the synthesis of the organophosphate-type NA, the -OH group of the acid part of the commercially available type NA-21 was substituted by an with isopropoxy group. The structure of APAl-3C was analyzed by spectroscopy and element analysis, the results of which were consistent with the theoretical molecular formula. The NA’s thermal stability was studied by differential scanning calorimetry (DSC) and thermogravimetry (TG), which showed only weak mass loss below 230 °C, meaning that it would not decompose during the processing of PP. The Li-salt of  APAl-3C (APAl-3C-12Li ) was used as a novel nucleating agent, studying its effects on crystallization, microstructure, mechanical and optical properties. Tests were performed in an iPP random copolymer at different contents, in comparison to the commercial NA-21. The composite with 0.5 wt% APAl-3C-12Li has a similar DSC  crystallization temperature of 118.8 °C as with the addition of 0.5 wt.-% NA-21. An advantage is that the composite with the APAl-3C-12Li has a lower haze value of  9.3% than the counterpart with NA-21. This is due to the weaker polarity of APAl-3C-12Li after the introduction of methyl and better uniform dispersion in PP matrix, resulting stronger improvement of optical and mechanical properties."

Another very weak part is the introduction. No fundamental papers or reviews on iPP nucleation are cited, and generally everything before 2010 seems to be ignored. The fact that we are talking about alpha-nucleation with a particulate type should at least be mentioned. Regarding concentration effects of organophosphates, including the respective effect on optics, the paper of Zhang et al., JMS-B 2003 (see https://doi.org/10.1081/MB-120021575) should be considered, and the work of Horváth et al. (RSC Advances 2014, see https://doi.org/doi.org/10.1039/C4RA01917B) is a good example of polarity variation effects on nucleation efficiency. in turn, the value of several papers dealing with carbon fibers, chitosan coating etc. for the present paper is not obvious and these should be removed. 

Figure quality needs improvement, too. the text in Figure 5 is far too small to be properly legible, the SEM images of Figure 7 are nondescript and could be omitted (unless their quality is improved - hot xylene or permanganic etching to reveal crystal features might be a good idea), and Figures 8 and 9 are again too small. 

At this point a note on the use of post-comma positions seems justified as well: DSC temperature and enthalpy readings are never more precise than +/- 0.5 °C, so anything beyond the first post-comma position is superfluous. The same holds for haze and impact strength, and giving any post-comma positions for Flexural modulus is clearly inappropriate (ISO standard gives a precision of +/- 5%, which means 50 at 1000 MPa. 

Detail comments: 

- The  iPP used is obviously (from Tm in DSC) an ethylene-propylene random copolymer; some basic parameters like MFR and comonomer content should be given to allow comparability to other studies.

- page 4, line 165: better "... has often been applied ..."

- Description of sample preparation (by injection molding or compression?) for mechanical and optical testing needs to be specified (melt and mold temperature are required to value the relevance of the presented data).

- Why was a Vicat machine used for HDT? That's a different geometry!

- Table 2 needs to further horizontal lines to improve readability.

- A table with mechanical data would be helpful.

- The conclusions are very short and miss references to the previous knowledge.

Comments on the Quality of English Language

On top of all the details listed above, proper proof reading and correction of grammar / style glitches are clearly required.

Author Response

Dear Reviewer,

We are very grateful for your helpful comments and suggestions for revision (Paper ID: polymers-2699160). The manuscript has been revised carefully base on the comments of reviewers.

This is a very nice study of a novel organophosphate-type nucleating agent for iPP. The paper, however, suffers from a number of detail issues and mostly from poor language quality (see below).

1)As an example, please find here a suggested better version of the abstract (remember, abbreviations must be spelt out at their first use - and the abstract IS the first use of many!):

"In this work, a novel α-nucleating agent (NA) for isotactic polypropylene (iPP) termed APAl-3C-12Li was prepared and evaluated compared with the commercially available type NA-21. For the synthesis of the organophosphate-type NA (APAl-3C), the -OH group of the acid part of NA-21 was substituted by an with isopropoxy group. The structure of APAl-3C was analyzed by spectroscopy and element analysis, the results of which were consistent with the theoretical molecular formula. The APAl-3C’s thermal stability was studied by differential scanning calorimetry (DSC) and thermogravimetry (TG), which showed only weak mass loss below 230 °C, meaning that it would not decompose during the processing of PP. The APAl-3C-12Li was used as a novel nucleating agent, studying its effects on crystallization, microstructure, mechanical and optical properties. Tests were performed in an iPP random copolymer at different contents, in comparison to the commercial NA-21. The composite with 0.5 wt% APAl-3C-12Li has a similar crystallization temperature of 118.8 °C as with the addition of 0.5 wt% NA-21. An advantage is that the composite with the APAl-3C-12Li has a lower haze value of 9.3% than the counterpart with NA-21. This is due to the weaker polarity of APAl-3C-12Li after the introduction of methyl and better uniform dispersion in PP matrix, resulting stronger improvement of optical and mechanical properties."

Answer: Thank you for your recognition of this paper. We re-write the abstract. The added abstract is “In this work, a novel α-nucleating agent (NA) for polypropylene (PP) termed APAl-3C-12Li was prepared and evaluated compared with the commercially available type NA-21. For the synthesis of the organophosphate-type NA (APAl-3C), the -OH group of the acid part of NA-21 was substituted by an with isopropoxy group. The structure of APAl-3C was analyzed by spectroscopy and element analysis, the results of which were consistent with the theoretical molecular formula. The APAl-3C’s thermal stability was studied by differential scanning calorimetry (DSC) and thermogravimetry (TG), which showed only weak mass loss below 230 °C, meaning that it would not decompose during the processing of PP. The APAl-3C-12Li was used as a novel nucleating agent, studying its effects on crystallization, microstructure, mechanical and optical properties. Tests were performed in an iPP random copolymer at different contents, in comparison to the commercial NA-21. The composite with 0.5 wt% APAl-3C-12Li has a similar crystallization temperature of 118.8 °C as with the addition of 0.5 wt% NA-21. An advantage is that the composite with the APAl-3C-12Li has a lower haze value of 9.3% than the counterpart with NA-21. This is due to the weaker polarity of APAl-3C-12Li after the introduction of methyl and better uniform dispersion in PP matrix, resulting stronger improvement of optical and mechanical properties.”

  • Another very weak part is the introduction. No fundamental papers or reviews on iPP nucleation are cited, and generally everything before 2010 seems to be ignored. The fact that we are talking about alpha-nucleation with a particulate type should at least be mentioned. Regarding concentration effects of organophosphates, including the respective effect on optics, the paper of Zhang et al., JMS-B 2003 (see https://doi.org/10.1081/MB-120021575) should be considered, and the work of Horváth et al. (RSC Advances 2014, see https://doi.org/10.1039/C4RA01917B) is a good example of polarity variation effects on nucleation efficiency. in turn, the value of several papers dealing with carbon fibers, chitosan coating etc. for the present paper is not obvious and these should be removed.

Answer: Thank you for your instructive advice. We added the part about PP nucleation and alpha-nucleation. We added the two references to replace the references with carbon fibers, chitosan coating.

3)Figure quality needs improvement, too. the text in Figure 5 is far too small to be properly legible, the SEM images of Figure 7 are nondescript and could be omitted (unless their quality is improved - hot xylene or permanganic etching to reveal crystal features might be a good idea), and Figures 8 and 9 are again too small.

Answer: Thank you for your instructive advice. The text in Figure 5 was enlarged and Figure 7 was omitted. Figures 8 and 9 was enlarged, too.

4)At this point a note on the use of post-comma positions seems justified as well: DSC temperature and enthalpy readings are never more precise than +/- 0.5 °C, so anything beyond the first post-comma position is superfluous. The same holds for haze and impact strength, and giving any post-comma positions for Flexural modulus is clearly inappropriate (ISO standard gives a precision of +/- 5%, which means 50 at 1000 MPa.

Answer: Thank you for your instructive advice. The use of post-comma positions in this paper were modified.

5)Detail comments:

- The iPP used is obviously (from Tm in DSC) an ethylene-propylene random copolymer; some basic parameters like MFR and comonomer content should be given to allow comparability to other studies.

Answer: Thank you for your instructive advice. In this paper, We used PP (random copolymer) as matrix, not iPP (isotactic polypropylene).

- page 4, line 165: better "... has often been applied ..."

Answer: Thank you for your instructive advice. We have corrected the part.

- Description of sample preparation (by injection molding or compression?) for mechanical and optical testing needs to be specified (melt and mold temperature are required to value the relevance of the presented data).

Answer: Thank you for your instructive advice. The sample preparation (by injection molding) for mechanical and optical testing was described in the part of 2.4, and the mold temperature were presented.

- Why was a Vicat machine used for HDT? That's a different geometry!

Answer: Thank you for your instructive advice. It is my fault. We used VST (Vicat softening temperature) replace HDT.

- Table 2 needs to further horizontal lines to improve readability.

Answer: Thank you for your instructive advice. The Table 2 and Table 3 with DSC data were showed in this paper.

- A table with mechanical data would be helpful.

Answer: Thank you for your instructive advice. The Table 4 and Table 5 with mechanical data were showed in this paper.

- The conclusions are very short and miss references to the previous knowledge.

Answer: Thank you for your instructive advice. The added conclusion was “The results of the SEM, FTIR and 1H-NMR indicated that the new nucleating agent APAl-3C-12Li with 5.56 wt% of P and 2.51 wt% of Al has been successfully synthesized. The prepared APAl-3C-12Li exhibits good thermal stability and causes the improvement in the mechanical and crystallization property, which is suitable for PP processing and using application. The nucleation ability of APAl-3C-12Li nucleating agents on PP is analyzed by DSC and POM compared with that of NA-21. Incorporating the nucleating agent APAl-3C-12Li into PP matrix can increase the crystallization temperatures and accelerate the crystallization rate of PP, indicating that the degree of ordering of the spherulites and the density of the spherulites are improved after refinement. The ΔHc value of the PP/AP composits with APAl-3C-12Li is always higher than that of PP/NA composits with NA-21 under the same addition amount, suggesting the superior nucleating ability of APAl-3C-12Li, which accelerates rapidly the crystallization of PP, which can be explained by the introduction of non-polar functional groups in APAl-3C-12Li structure. The mechanical analysis shows that the PP/AP composites has remarkable enhancement in terms of the tensile strength, the impact strength and the flexural modulus compared with that of pure PP. The presence of non-polar functional groups in APAl-3C-12Li structure promote the compatibility with PP matrix, which endows the PP/AP composits preferable impact in comparation with PP/NA composits. Furthermore, the noteworthy fact is that the PP/AP-3 composite with 0.5 wt% of APAl-3C-12Li has a lowest haze value of 9.3%, which is prior of that of PP/NA-3 composite with 0.5wt% of NA-21.”.

Reviewer 2 Report

Comments and Suggestions for Authors

The paper titled "Exploitation of a New Nucleating Agent by Molecular Structure  Modification of Aryl Phosphate and its Effect on Application Properties of the Polypropylene" presents a novel nucleating agent designed for polypropylene composites. This agent significantly improves various composite parameters by enhancing nucleation effects in the crystallization process and ensuring a more homogeneous distribution of APAl-3C within the polymer matrix. The paper is well-prepared for publication, pending a few minor revisions.

To enhance clarity, please correct the caption for Figure 4 to specify TG and DTG curves for 4a and DSC curves for 4b. Additionally, including TG, DTG, and DSC curves for NA-21 would be valuable for comparative analysis.

I recommend calculating wetting contact angles for both samples and incorporating this data into the text.

Furthermore, it would be beneficial to include a thorough discussion on the advantages and disadvantages of using APAl-3C in polypropylene composites when compared to similar substances.

Lastly, I suggest citing the highly relevant paper with the following https://doi.org/10.1002/masy.200450638, which explores the use of heterofunctional polyperoxides to enhance the properties of polypropylene composites.

Comments on the Quality of English Language

Minor editing of English language required.

Author Response

Dear Reviewer,

We are very grateful for your helpful comments and suggestions for revision (Paper ID: polymers-2699160). The manuscript has been revised carefully base on the comments of reviewers.

The paper titled "Exploitation of a New Nucleating Agent by Molecular Structure  Modification of Aryl Phosphate and its Effect on Application Properties of the Polypropylene" presents a novel nucleating agent designed for polypropylene composites. This agent significantly improves various composite parameters by enhancing nucleation effects in the crystallization process and ensuring a more homogeneous distribution of APAl-3C within the polymer matrix. The paper is well-prepared for publication, pending a few minor revisions.

Answer: Thank you for your recognition of this paper.

1)To enhance clarity, please correct the caption for Figure 4 to specify TG and DTG curves for 4a and DSC curves for 4b. Additionally, including TG, DTG, and DSC curves for NA-21 would be valuable for comparative analysis.

Answer: Thank you for your instructive advice. We have corrected the caption for Figure 4 to specify TG and DTG curves for 4a and DSC curves for 4b. Moreover, we evaluated the nucleation property and application property of APAl-3C-12Li (composed of APAl-3C and lithium laurate 12Li) compared with that of NA-21 (composed of APAl-OH and lithium laurate 12Li), so I think it is unnecessary to add TG, DTG, and DSC curves for NA-21 for comparative analysis with APAl-3C.

2)I recommend calculating wetting contact angles for both samples and incorporating this data into the text.

Answer: Thank you for your instructive advice. We have calculated the wetting contact angles for samples in this paper. The added part is “The hydrophobicity of APAl-3C is 89.7 °, higher than 77.3 ° of APAl-OH because of the presence of -CH3.”.

3)Furthermore, it would be beneficial to include a thorough discussion on the advantages and disadvantages of using APAl-3C in polypropylene composites when compared to similar substances.

Answer: Thank you for your instructive advice. We have added the the advantages and disadvantages of using APAl-3C in polypropylene composites when compared to similar substances in conclusion. The added conclusion was “The results of the SEM, FTIR and 1H-NMR indicated that the new nucleating agent APAl-3C-12Li with 5.56 wt% of P and 2.51 wt% of Al has been successfully synthesized. The prepared APAl-3C-12Li exhibits good thermal stability and causes the improvement in the mechanical and crystallization property, which is suitable for PP processing and using application. The nucleation ability of APAl-3C-12Li nucleating agents on PP is analyzed by DSC and POM compared with that of NA-21. Incorporating the nucleating agent APAl-3C-12Li into PP matrix can increase the crystallization temperatures and accelerate the crystallization rate of PP, indicating that the degree of ordering of the spherulites and the density of the spherulites are improved after refinement. The ΔHc value of the PP/AP composits with APAl-3C-12Li is always higher than that of PP/NA composits with NA-21 under the same addition amount, suggesting the superior nucleating ability of APAl-3C-12Li, which accelerates rapidly the crystallization of PP, which can be explained by the introduction of non-polar functional groups in APAl-3C-12Li structure. The mechanical analysis shows that the PP/AP composites has remarkable enhancement in terms of the tensile strength, the impact strength and the flexural modulus compared with that of pure PP. The presence of non-polar functional groups in APAl-3C-12Li structure promote the compatibility with PP matrix, which endows the PP/AP composits preferable impact in comparation with PP/NA composits. Furthermore, the noteworthy fact is that the PP/AP-3 composite with 0.5 wt% of APAl-3C-12Li has a lowest haze value of 9.3%, which is prior of that of PP/NA-3 composite with 0.5wt% of NA-21.”.

4)Lastly, I suggest citing the highly relevant paper with the following https://doi.org/10.1002/masy.200450638, which explores the use of heterofunctional polyperoxides to enhance the properties of polypropylene composites.

Answer: Thank you for your instructive advice. We have added the reference in this paper.

Round 2

Reviewer 1 Report

Comments and Suggestions for Authors

The paper is fine for publication with the revisions made.

Author Response

Dear reviewer:

We are very grateful for your helpful suggestions for revision (Paper ID: polymers-2699160). Thank you for your arduous work.

Thank you very much!

Best regards,

Fu-hua Lin